# Maleimide-Functionalized Liposomes: Prolonged Retention and Enhanced Efficacy of Doxorubicin in Breast Cancer with Low Systemic Toxicity

**DOI:** 10.3390/molecules27144632

**Published:** 2022-07-20

**Authors:** Chuane Tang, Dan Yin, Tianya Liu, Rui Gou, Jiao Fu, Qi Tang, Yao Wang, Liang Zou, Hanmei Li

**Affiliations:** 1Key Laboratory of Coarse Cereal Processing, School of Food and Biological Engineering, Chengdu University, Chengdu 610106, China; 18780753949@163.com (C.T.); enidyin@outlook.com (D.Y.); 18980184237m@sina.cn (J.F.); 2School of Mechanical Engineering, Chengdu University, Chengdu 610106, China; tianyaliu88@gmail.com (T.L.); gouruiii@163.com (R.G.); 3Sichuan Industrial Institute of Antibiotics, School of Pharmacy, Chengdu University, Chengdu 610106, China; tangqq2022@163.com (Q.T.); 18328750510@163.com (Y.W.)

**Keywords:** breast cancer, liposome, maleimide, doxorubicin, sustained release

## Abstract

Cell surface thiols can be targeted by thiol-reactive groups of various materials such as peptides, nanoparticles, and polymers. Here, we used the maleimide group, which can rapidly and covalently conjugate with thiol groups, to prepare surface-modified liposomes (M-Lip) that prolong retention of doxorubicin (Dox) at tumor sites, enhancing its efficacy. Surface modification with the maleimide moiety had no effect on the drug loading efficiency or drug release properties. Compared to unmodified Lip/Dox, M-Lip/Dox was retained longer at the tumor site, it was taken up by 4T1 cells to a significantly greater extent, and exhibited stronger inhibitory effect against 4T1 cells. The in vivo imaging results showed that the retention time of M-Lip at the tumor was significantly longer than that of Lip. In addition, M-Lip/Dox also showed significantly higher anticancer efficacy and lower cardiotoxicity than Lip/Dox in mice bearing 4T1 tumor xenografts. Thus, the modification strategy with maleimide may be useful for achieving higher efficient liposome for tumor therapy.

## 1. Introduction

Cancer is the second leading cause of death worldwide [1], accounting for 1/6 of global deaths [2]. Among the different types of cancer, breast cancer has the second highest incidence and is the leading cause of cancer-related death in women [3]. The main methods for treating cancer are chemotherapy, radiotherapy, immunotherapy, and surgery [4,5].

Doxorubicin (Dox) is a chemotherapeutic drug frequently used to treat several types of cancer, including cancers of the breast, ovaries, lungs, and bladder, as well as multiple myeloma [6]. Although various Dox-based formulations have been developed for clinical use, such as Doxil^®^, Doxil/Caelyx, and Lipodox [7], high Dox concentrations can lead to serious side effects [8] and systemic toxicity in the heart, liver, and kidney [9]. Dox-induced cardiomyopathy is the most common adverse event, affecting around 11% of patients; this complication is associated with poor prognosis, including a mortality rate of up to 50% if it progresses to congestive heart disease [10]. Multidrug resistance in malignant tumors [11], arising when P-glycoprotein in tumors efficiently pumps out the drug [12], also limits the clinical application of Dox, especially after long-term use of high doses. Therefore, novel drug delivery systems are required to achieve sustained release at the tumor site. Since intratumoral injection can reduce toxicity and improve drug efficiency [13], we assumed that the in situ injection of sustained-release formulations may prolong the retention of Dox in the tumor and reduce damage to other organs.

Liposomes are widely used as drug carriers in the clinic [14], and several liposome-based Dox formulations, such as Doxil/Caelyx and Lipodox, have been approved for the treatment of metastatic breast cancer and Kaposi’s sarcoma [7,15]. Indeed, liposomes are currently considered the ideal delivery platform for Dox because they are biocompatible and easy to prepare [16], and because their injection into the tumor offers controlled, sustained drug release [17]. Nevertheless, new liposome formulations are needed that are retained longer in tumors, that release drug more slowly, and that are more acceptable to patients.

Maleimide is a stable, easy-to-use functional group that can covalently react via Michael addition with the mercaptan group of cysteine residues in proteins and peptides [18]. In recent years, reactive mercaptan groups on the cell surface have been identified as a new target for promoting the adsorption and subsequent endocytosis of maleimide-modified nanoparticles, peptides, oligonucleotides, polymers, and dyes [19]. For instance, maleimide-functionalized polymers have shown excellent adhesion properties to porcine bladder mucosa [20] due to their ability to form covalent bonds with the sulfhydryl groups in mucins [21]. Nevertheless, the intratumoral injection of maleimide-functionalized liposomes for cancer treatment has not yet been reported.

Here, we prepared maleimide-functionalized Dox-loaded liposomes (M-Lip/Dox) and we evaluated their biocompatibility, uptake by tumor cells and retention in tumors, drug release, and antitumor efficacy (Figure 1). We expect that our results will serve as a guide for the development of novel sustained-release formulations for cancer treatment.

## 2. Materials and Methods

### 2.1. Materials

Distearoyl phosphatidylethanolamine-maleimide (DSPE-Mal) was obtained from Xian Ruixi Biotechnology (Xi’an, China); lipoid S100, from AVT Pharmaceutical (Shanghai, China); cholesterol, from Kelong Chemical Company (Chengdu, China); 3-(4,5-dimethyl-2-thiazolyl)-2,5-diphenyl-2*H*-tetrazolium bromide (MTT), from Sigma-Aldrich (St. Louis, MO, USA); and trypsin, penicillin, and streptomycin, from Dongheng Huadao Biotechnology (Shanghai, China). Dox was provided by Dalian Meilun Biotechnology, and 1,1′-dioctadecyl-3,3′,3,3′-tetramethyl indodicarbocyanine (DiD) by Biotium (Hayward, CA, USA). All other chemicals and reagents were of analytical grade and were commercially available.

### 2.2. Cell Culture and Animals

Mouse breast cancer (4T1) cell lines were obtained from the Shanghai Institutes for Biological Sciences (Shanghai, China). Cells were maintained in Dulbecco’s modified Eagle’s medium (DMEM) supplemented with 10% fetal bovine serum (FBS) and 1% penicillin/streptomycin, and grown at 37 °C in an atmosphere containing 5% CO_2_. Cultures were passaged using 0.1% trypsin-ethylenediaminetetraacetic acid.

BALB/c mice (17–19 g, male) were obtained from Beijing Huafukang Biotechnology. All animal experiments were performed in accordance with the US National Institutes of Health “Guidelines for Care and Use of Laboratory Animals”, and were approved by the Experimental Animal Management Committee of Chengdu University.

### 2.3. Liposome Preparation

Dox-loaded liposomes (Lip/Dox) were prepared by a thin-film hydration method [22]. Lipoid S100 and cholesterol were first dissolved in chloroform at a mass ratio of 4:1. Evaporation of chloroform at 37 °C afforded a thin film, which was then hydrated with 123 mM ammonium sulfate and probe-sonicated at 195 W for 5 min. The obtained preparation was passed through a G-25 dextran gel column (Solarbio, Beijing, China) and incubated with Dox at 55 °C for 20 min to give Lip/Dox.

M-Lip/Dox was prepared by the same method using lipoid S100, cholesterol, and DSPE-Mal at a mass ratio of 14:6:5. For DiD-labeled liposomes, an appropriate amount of DiD was added to the lipid solution in chloroform prior to evaporation.

### 2.4. Liposome Characterization

The size, zeta potential, and polydispersity index of the prepared liposomes were determined by dynamic light scattering (Zen3600, Malvern Instruments, Malvern, UK), and their stability was assessed by observing the change in particle size during storage at 4 °C during one week. All samples were measured in triplicate.

Liposome morphology was observed by transmission electron microscopy (TEM; H-600, Hitachi, Tokyo, Japan). Briefly, liposomes were diluted into deionized water, placed on a copper mesh covered with carbon film, and then negatively stained with 2.0% sodium phosphotungstate solution at room temperature for 30 s.

### 2.5. Drug Loading and Encapsulation Efficiencies

Lip/Dox and M-Lip/Dox were passed through an equilibrated sepharose G-25 column to remove free Dox, and their drug loading efficiency (DL%) was measured using a fluorescence spectrometer (FL970, Techcomp, Beijing, China) at an excitation wavelength of 485 nm and emission wavelength of 590 nm. The DL% was measured according to following Equation (1):(1)$$\text{DL}\%\;=\frac{\text{Weight of drug in liposomes}}{\text{Total weight of liposomes}}\times 100\%$$

Encapsulation efficiency (EE%) was measured using the same fluorescence procedure and calculated according to the following Equation (2):(2)$$\text{EE}\%\;=\frac{\text{Weight of drug in liposomes}}{\text{weight of drug added}}\times 100\%$$

### 2.6. In Vitro Drug Release

The release of Dox from Lip/Dox and M-Lip/Dox was measured in vitro using a dialysis method [23,24]. First, 0.2 mL of each sample was placed into a dialysis bag with a molecular weight cut-off value of 8–14 kDa. Then, the dialysis bag was immersed in 19.8 mL of phosphate-buffered saline (PBS, pH 6.8) and incubated at 37 °C with continuous shaking at 100 rpm. At scheduled time points (0, 2, 4, 8, 12, 24, 48, and 72 h), 1 mL aliquots were collected from the dialysate and replaced with the same volume of fresh PBS. Drug release was determined using the FL970 fluorescence spectrometer at 488 nm, and the cumulative drug release was calculated as follows (Equation (3)):(3)$$\text{Cumulative drug release}\;\left(\%\right)=\frac{M_{t}}{M_{0}}\times 100\%$$
where M_t_ is the amount of drug released at time t and M_0_ is the amount of drug initially loaded into liposomes. The drug concentration at each sampling point was calculated based on a calibration curve obtained using solutions of known Dox concentration.

### 2.7. Liposome Uptake by Cancer Cells

4T1 cells were seeded in six-well plates (1 × 10^6^ cells/well), incubated with free Dox, Lip, or M-Lip at 37 °C for 1.5 h, then the cells were washed with cold PBS, detached with trypsin, and centrifuged three times at 120 g for 5 min. The uptake of all formulations was measured using a flow cytometer (FACSCanto^TM^ II, BD, NJ, Paterson, USA). All preparations were tested in triplicate.

### 2.8. Cytotoxicity of Blank Liposomes

4T1 cells were seeded in 96-well plates (1 × 10^4^ cells/well) and incubated in a 5% CO_2_ atmosphere at 37 °C for 12 h. The culture medium was then replaced with 100 µL of fresh DMEM containing different concentrations of blank Lip or blank M-Lip (10, 20, 50, 100 µg/mL). After incubation at 37 °C for 24 h, MTT (20 µL, 5 mg/mL) was added to each well, followed by incubation for another 4 h. The culture medium was then removed and 150 μL of DMSO was added. After air-bath (constant temperature shaker, CHA-S, Shanghai, China) shaking at 37 °C for 10 min, the absorbance at 570 nm was measured (Varioskan Flash, Thermo, MA, USA) and cell viability was determined according to the following Equation (4):(4)$$\text{Cell viability}\;\left(\%\right)=\frac{A_{s}}{A_{c}}\times 100\%$$
where A_c_ is the absorbance of cells treated with fresh medium and A_s_ is the absorbance of cells treated with blank Lip or M-Lip.

### 2.9. In Vitro Antitumor Efficacy of Dox-Loaded Liposomes

4T1 cells were seeded in 96-well plates (1 × 10^4^ cells/well) and incubated at 37 °C for 12 h. Then, the culture medium was replaced with 100 μL of fresh DMEM containing free Dox, Lip/Dox, or M-Lip/Dox with various Dox concentrations (1.0, 2.0, 5.0, 10.0 µg/mL). The cells were then incubated for another 24 h and the antitumor efficacy of Dox-loaded liposomes was assessed using the MTT assay as described above.

### 2.10. Tumor-Bearing Mouse Model

To establish the tumor-bearing mouse model, 5 × 10^6^ 4T1 cells suspended in 100 µL of DMEM were injected into the second right mammary fat pad of mice [25]. The experiments complied with the principles of Compound Laboratory Animal Protection.

### 2.11. Near-Infrared Fluorescence Imaging

To study the retention time of drug-loaded liposomes in vivo, mice were intratumorally injected with DiD-labeled liposomes (10 µg/mouse). At 0.5, 2, 4, 6, 24, 48, and 72 h post-injection, mice were imaged using a Kodak FX PRO in vivo imaging system (Carestream Health, Rochester, NY, USA) equipped with a 475 nm excitation band-pass filter and set to an emission wavelength of 575 nm. The exposure time was 20 s per image.

### 2.12. Antitumor Efficacy and Safety In Vivo

4T1 cells (5 × 10^6^) were subcutaneously injected into the right flank of each mouse. At seven days post-injection, mice were randomly divided into four groups (n = 7): control, free Dox, Lip/Dox, and M-Lip/Dox. Mice in the control group were intratumorally injected with 100 µL of saline, while mice in the other three groups were intratumorally injected with 5 mg/kg of Dox. The tumor volume and weight in all groups were monitored for 18 days. The tumor volume was measured using a digital caliper and calculated as follows Equation (5):(5)$$\text{Tumor Volume}\;=\frac{\text{Length}\times {\text{Width}}^{2}}{2}$$

On day 18, three mice from each group were euthanized, and their heart and tumor were harvested, fixed, and embedded in paraffin. Mice were excluded if the tumor volume was greater than 1000 mm^3^, if weight loss > 20%, or if signs of weakness were observed. The collected tumor tissues were then sectioned for necrosis assay by the terminal deoxynucleotidyl transferase dUTP-biotin nick-end labeling (TUNEL) assay. The cell nuclei were stained with 4′-diamino-2-phenylindole and the number of apoptotic cells was determined with a fluorescence microscope (Nikon, Eclipse Ci-L, Tokyo, Japan). The collected heart tissues were sectioned and assayed by hematoxylin-eosin staining.

### 2.13. Statistical Analysis

Statistical analysis was performed using a two-tailed *t*-test or analysis of variance. All data were presented as mean ± standard deviation (SD). Differences associated with *p* < 0.01 were considered statistically significant.

## 3. Results and Discussion

### 3.1. Liposome Characterization

Liposome size is known to affect the stability, uptake, and metabolism of liposome-based formulations [26]. We found that Lip and M-Lip were spherical nanoparticles with a similar size of ~100 nm (Figure 1A,C). As shown in the Figure 1B, the zeta potential of M-Lip/Dox −39 mV, which is significantly lower than Lip/DOX (−22.2 mV). The particle size of both Lip/Dox and M-Lip/Dox did not change after storage at 4 °C for one week (Figure 1D), indicating that the prepared formulations were stable for further experiments. Moreover, the encapsulation and drug loading efficiencies of M-Lip/Dox were estimated at 98% and 10%, respectively, suggesting that the maleimide modification liposome has good drug loading performance.

Slow release is an important requirement for injectable drugs, as it helps to reduce the frequency of administration, maintain high local concentrations of the drug, and improve drug efficacy [27]. Here, we found that DOX release from Lip/Dox and M-Lip/Dox was characterized with an initial burst release, reaching 30% within the first 6 h (Figure 1E). Then, the release became much slower and reached 35% at 72 h. However, the drug release of free Dox went up to 60% within the first 2 h, indicating that loading of Dox into liposomes could lead to significantly better stability and slowed the release rate of drugs. In addition, M-Lip/Dox possessed a similar drug release profile with Lip/Dox, indicating that the modification of maleimide moiety had no effect on the drug release rate of liposomes.

### 3.2. Uptake by Cells and Cytotoxicity

The uptake of the liposome formulations into 4T1 cells was studied by flow cytometry. The results showed that the fluorescence intensity of M-Lip/Dox was stronger than that of Lip/Dox (Figure 2A), indicating that the maleimide modification can enhance the uptake of liposomes into cells.

Dox is a low-molecular-weight compound that can enter nuclei and bind to DNA to induce apoptosis [28]. Therefore, the drug delivery efficacy of the prepared liposomes was assessed in terms of Dox-induced cytotoxicity [29]. We found that blank Lip and blank M-Lip were non-toxic at low concentrations (Figure 2B). However, M-Lip/Dox effectively inhibited cell growth in a dose-dependent manner, showing a stronger inhibitory effect than Lip/Dox (Figure 2C). This suggests that maleimide functionalization can improve the uptake of liposomes into cells and enhance the antitumor efficacy of Dox.

### 3.3. Liposome Retention in the Tumor

Longer retention of liposomes can prevent their degradation in circulation, minimize their toxicity, enhance their pharmacokinetic profile, and lead to improved patient compliance [30,31,32]. As such, the retention time of the liposome at the tumor site was monitored. 4T1 tumor-bearing BALB/c mice were injected intratumorally with DiD-labeled Lip/Dox or M-Lip/Dox and imaged at scheduled time points by near-infrared fluorescence imaging. Although both Lip and Mal-Lip accumulated at the tumor site, the fluorescence intensity in the M-Lip/Dox group was much stronger than that in the Lip/Dox group at 72 h post-injection (Figure 3A). As shown in the Figure 3B, the fluorescence intensity of both Lip/Dox and M-Lip/Dox increased in the first 6 h, and decreased after 24 h. In addition, the fluorescence intensity of M-Lip/Dox decreased more slowly than Lip/Dox. This suggests that the maleimide modification can greatly improve the retention of liposomes in the tumor.

### 3.4. In Vivo Antitumor Efficacy and Safety

The antitumor efficacy of Lip/Dox and M-Lip/Dox in vivo was evaluated using tumor-bearing mice. Tumors grew rapidly in mice treated with PBS, but a slower trend was observed in mice treated with free Dox, Lip/Dox, or M-Lip/Dox (Figure 4A). The tumor inhibition effect of M-Lip/Dox group was the strongest among all the groups. Nevertheless, none of the formulations caused significant changes in body weight (Figure 4B), suggesting that both Lip/Dox and M-Lip/Dox are safe for in vivo application.

On day 18, tumor tissues were collected from each treatment group and cell apoptosis was studied by TUNEL staining. Strong green fluorescence was observed in the M-Lip/Dox group, followed by the Lip/Dox and free Dox groups (Figure 4C), suggesting that M-Lip/Dox had the strongest ability to induce tumor apoptosis. This may because M-Lip/Dox can enter tumor cells more efficiently and be retained in tumor tissue longer than non-functionalized formulations. The improvement in cell internalization and drug retention time is believed to be due to the contribution of the thiol-reactive maleimide-modified surface [17].

Potential Dox-induced cardiac injury was also studied by histological analysis (Figure 4D). The epicardium of the right ventricle in animals treated with free Dox showed mild edema, loose structure, scattered inflammatory cell infiltration, mild myocardial vacuolar degeneration, and small vacuoles in the cytoplasm. Animals treated with Lip/Dox or M-Lip/Dox showed normal structure of the endo, myo, and epicardium, and no obvious abnormality in the cardiac cavity. Thus, compared with free Dox and Lip/Dox, M-Lip/Dox may be an efficient and safe drug delivery system.

## 4. Conclusions

We prepared maleimide-functionalized liposomes to improve the antitumor efficacy and retention time of Dox at the tumor site. Compared to free Dox or non-functionalized liposomes, M-Lip/Dox was taken up more efficiently by 4T1 cells, it was retained longer in xenografts in vivo, and it showed stronger antitumor effects in vivo. At the same time, animals treated with M-Lip/Dox showed negligible cardiac toxicity, in contrast to the substantial injury caused by free Dox. Thus, surface modification of liposomes with the maleimide group may serve as a new strategy for the development of sustained-release liposomes to deliver anticancer treatment.

## Data Availability

Not applicable.

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
