# Peer review of "Maleimide-Functionalized Liposomes: Prolonged Retention and Enhanced Efficacy of Doxorubicin in Breast Cancer with Low Systemic Toxicity"

_molecules, 2022, doi:10.3390/molecules27144632_

Round 1

Reviewer 1 Report

- The Introduction section must be extended with latest papers related with Doxorubicin drug delivery behavior (release from protein nanocarries, synthetic nanocarriers, etc.): J Nanomed Nanotechnol. 2018; 9(5): 519;  Polymers, 2021, 13(13), 2047,  etc.

- The release test revelead a burst release in the first hours (probably 6-8 hours). Here, the majority of the drug amount was released. Thus, the second release stage (until 72 hours) cannot be exhibit as a sustained/prolonged release. Please rephrase.

- Please extend the discussion related with the differences (in terms of release behavior, efficiency) between the formulations.

- Please provide a proof of the liposomes surface modification (FTIR/RAMAN/RMN/XPS)

- Please extend the discussion related with the Zeta and DLS results. I think that there are some issues about liposomes agglomeration. These issues are directly related with the Zeta potential values. You have to check again the Zeta potential analysis to have reliable results. It is hard to belive that a surface modification does not affect the surface charging.

Author Response

Thanks for your suggestion.  We have corrected the manuscript as the suggestion carfully. 

Reviewer 2 Report

The paper is well written, good English, well described methods. There are some remarks

Methods

1. TUNEL description is lost

2. Injection of liposomes for atitumor activity estimation is not stated (iv or it?)

Results

1. Bioimaging do not seem statistically significant, please use ImageJ or alike programm to calculate MFI

2. In vivo results do not seem reliable. Free DOX and Lip/DOX demonstrated the same antitumor activity in vivo which is suspitious.

3. Statistical differences in some figures do not seem significant.

4. Heart histology do not show any difference between  Lip/DOX and M-Lip/DOX, should be removed or conclusions corrected.  

Please see remarks in the text

Author Response

Thanks for you suggestion.  We have revised the manuscript according your suggestion carfully.

Round 2

Reviewer 2 Report

All the remarks were met.